# Are there Metacognitivists in the Fox Hole? A Preliminary Test of Information Seeking in an Arctic Fox (*Vulpes lagopus*)

**DOI:** 10.3390/bs10050081

**Published:** 2020-04-26

**Authors:** Taryn Eaton, Patricia Billette, Jennifer Vonk

**Affiliations:** 1Psychology Department, Oakland University, Rochester, MI 48309, USA; tarynmeaton@gmail.com; 2Animal Care, The Creature Conservancy, Ann Arbor, MI 48103, USA; patricia.billette@gmail.com

**Keywords:** arctic fox, information seeking, metacognition, carnivore

## Abstract

Over the last two decades, evidence has accrued that at least some nonhuman animals possess metacognitive abilities. However, of the carnivores, only domestic dogs have been tested. Although rarely represented in the psychological literature, foxes are good candidates for metacognition given that they cache their food. Two experiments assessed metacognition in one male arctic fox (*Vulpes lagopus*) for the first time. An information-seeking paradigm was used, in which the subject had the opportunity to discover which compartment was baited before making a choice by looking through a transparent window in the apparatus. In the first experiment, choice accuracy during seen trials was equal to choice accuracy on unseen trials. Importantly, there was no significant difference between the subject’s looking behavior on seen versus unseen trials. In the second experiment, with chance probabilities reduced, the subject’s choice accuracy on both seen and unseen trials was below chance. The subject did not exhibit looking behavior in any of the trials. Latencies to choose were not influenced by whether he witnessed baiting. Although we did not obtain evidence of metacognition in our tests of a single subject, we maintain that foxes may be good candidates for further tests using similar methodologies to those introduced here.

## 1. Introduction

Metacognition, or knowing what one knows, involves two distinct processes: monitoring and control [1]. Monitoring refers to making judgements about one’s memory and/or cognition, whereas control involves using metacognitive judgements to guide one’s behavior [2]. For example, when studying for an exam, students make judgements about how much they have learned by assessing their current state of knowledge (monitoring). These judgements are used to stop studying when students feel they have adequately learned the material (control). Metacognition was once thought to be unique to humans [2]. However, evidence in the last few decades suggests that some nonhumans may possess metacognitive abilities (e.g., dolphins [3], rats [4], rhesus macaques [5], chimpanzees and orangutans [6], and western scrub jays, [7]; see reviews in [8,9]). Given that arctic foxes bury their food for future need (i.e., caching), which involves remembering the location of cached food, they are good candidates for metacognitive abilities. However, no fox species have been tested to our knowledge. Here, we develop two procedures appropriate for testing information seeking in an arctic fox for the first time.

Initial studies addressed the ability to monitor knowledge by testing uncertainty monitoring (the capacity of a person or animal to recognize when they are lacking confidence about a relevant piece of information; [3,10,11]). Many uncertainty monitoring tasks use choice discriminations, such as distinguishing long tones from short tones, of varying discriminability. Subjects are then given an opportunity to emit an uncertain response, (e.g., selecting a specific button or lever) that allows them to opt out of trials and still receive a reward, albeit usually a smaller or less preferred reward than they would receive on correct trials [3]. The rationale for these studies is that an animal with metacognitive access (i.e., knowing when memory is likely to fail or performance is likely to be inaccurate) should opt out of more difficult trials, given that the ability to judge accurately and gain a reward is lower [12]. 

Smith and colleagues [3] were among the first to use such a paradigm. They trained a dolphin (*Tursiops truncatus*) to discriminate between tones ranging from 1200 to 2100 Hz by approaching the correct paddle based on its spatial location. For a 2100 Hz “high trial”, the left paddle was correct. For a “low trial” of 1200 to 2099 Hz, the right paddle was correct. When the dolphin chose the correct paddle, a food reward was given. The dolphin was also presented with a third “opt-out” paddle placed in between the left and right paddles. If the middle paddle was depressed, the ongoing trial was discontinued and replaced with a low trial that the dolphin always got correct. The dolphin’s responses were then compared to human responses on an analogous task. Smith et al. found that the dolphin was more likely to use the third “opt out” paddle when the tone thresholds were indistinct (e.g., 2099 Hz) compared to when the tone thresholds were clearly distinguishable. 

They also found that the dolphin’s behavior (e.g., slowed approach to the paddles, wavered among paddles, swam toward the paddles with an open mouth while sweeping his head from side to side, or while opening and closing his mouth rhythmically) when the tone thresholds were indistinct reflected apparent uncertainty. Behavior reflecting apparent uncertainty has also been identified in rhesus macaques (*Macaca mulatta*) during similar judgement tasks [13]. Foote and Crystal [4] found that, when given the option, lab rats (*Rattus norvegicus*) chose to opt out of a noise length discrimination task at higher rates as the difficulty of discrimination increased. Furthermore, Beran and colleagues [5] found that rhesus macaques declined to make numerosity judgments (more or fewer dots than a center value) on trials in which the number of dots presented on a screen were closer to a center value, and thus, more difficult to discriminate. Thus, many distantly related species have shown some evidence of uncertainty monitoring. 

Modified uncertainty paradigms have also been used to test metamemory, or the knowledge of what one does and does not remember [10]. Kornell, Son, and Terrace [14] presented rhesus monkeys with six sample pictures on a touch screen. Nine pictures were then shown simultaneously—one of which had been presented among the previous sample pictures. Subjects were required to touch the sample picture and then choose an icon based on how confident they were about their responses. If the high confidence icon was chosen, a correct response to the memory task resulted in a large reward and an incorrect response resulted in no reward. If the low confidence icon was chosen, a small reward was given regardless of whether the response to the memory task was correct or incorrect. Rhesus monkeys chose the high-confidence icon more frequently after correct responses than after errors, and they chose the low-confidence icon more frequently after errors than after correct responses. Similarly, Hampton [15] had rhesus macaques perform a delayed matching-to-sample task. On some of the trials, the macaques were given an option to opt out of completing the task for a less preferred food reward. Hampton found that the macaques’ matching performance declined with longer delays between sample presentation and presentation of the match, that they declined memory tests at long retention intervals, and that they maintained strong performance levels even at long retention intervals when they were given the opportunity to choose when they would complete the memory test. This evidence suggests that rhesus macaques are able to monitor internal signals of remembering. Tufted capuchin monkeys (*Cebus apellaa*) and orangutans (*Pongo pygmaeus*) have also been shown to opt out of memory test trials in which they are likely to err [16,17]. However, Beran and colleagues [18] found that capuchin monkeys rarely or never chose to use an uncertainty response when given a psychophysical discrimination task (e.g., density discriminations), and metamemory tests conducted with pigeons (*Columba livid*) have produced null results [10]. Thus, although some components of metacognition appear broadly distributed in the animal kingdom, it is not ubiquitous.

The aforementioned studies suggest that some subjects knew when they did and did not know the correct response to a discrimination task. However, critics of uncertainty paradigms argue that subjects may not recognize their state of uncertainty but, instead, may learn to use the escape response in the presence of certain stimuli [19]. Subjects may form associations between observable cues and outcomes such as learning to avoid tests that include stimuli in a specific magnitude range after experiencing low rates of reward for those stimuli [8]. In Hampton’s study [15], the macaques may have learned to use the longer delay as a cue to opt out, given that opting out resulted in a better reward on longer-delay trials compared to shorter-delay trials, without necessarily understanding the implications for their own internal memory states. Subjects may also form behavioral cue associations. For example, subjects may hesitate when they do not know which response to make. During this period of hesitation, subjects may engage in other behaviors, such as looking back and forth between choice options. These behaviors then become associated with the escape response [8]. Whether uncertainty paradigms demonstrate metacognitive abilities as opposed to alternative mechanisms remains controversial [12].

More recently, researchers have focused on the second aspect of metacognition: the ability to control one’s knowledge state. Typically, in these studies, an animal is placed in a problem-solving situation in which a piece of information is missing, but may be gathered, to solve the problem successfully [12]. Call and Carpenter [6] developed an information-seeking paradigm to assess whether human children, chimpanzees (*Pan troglodytes*) and orangutans (*Pongo pygmaeus*) were capable of controlling their knowledge states. Subjects were presented with a set of opaque tubes in which food was hidden. The experiment consisted of two sets of trials, one set in which subjects witnessed the baiting of the tubes and another set in which the subjects did not witness the baiting. During testing, subjects could select one tube and collect the reward. When subjects had witnessed the baiting of the tubes, all subjects chose that tube almost immediately. When subjects had not witnessed the baiting, they looked down into the open ends of the tubes before choosing. This finding suggests that subjects knew when they did and did not know where the reward was. Watanabe and Clayton [7] used a similar information-seeking task to investigate metacognition in Western scrub jays (*Aphelocoma californica*). They created uncertainty about the food location by making the tube baiting process visibly unavailable by inserting a delay between baiting and food retrieval, and by moving the location of the bait. They found that the jays looked in the tubes more often during conditions that were consistent with high uncertainty.

The information-seeking paradigm has several advantages over other methods. Information-seeking studies do not require the extensive training required for many of the uncertainty monitoring studies [12]. Further, environmental cue and behavioral cue association explanations are less likely, as fewer critical test trials are required; thus, there is less time to learn the connections between cues and reward outcomes [8]. Results from information-seeking experiments are similarly unlikely to be accounted for by stimulus cues, because stimuli are indistinguishable and equally likely to be baited on any given trial [12]. Finally, in information-seeking experiments, an animal’s knowledge state is definitively known. The animal does or does not know where the food is, based on their visual access to the baiting procedure. This differs from uncertainty monitoring experiments, in which the ambiguity of discriminations varies and is used to predict trial difficulty [12].

Not all animals tested with the information-seeking paradigm are successful. Capuchin monkeys have repeatedly failed to respond accurately in experiments using this paradigm [20,21,22]. Dogs (*Canis familiaris*) also seem to perform inconsistently on information-seeking tasks. Bräuer, Call, and Tomasello [23] used the information-seeking paradigm to test metacognition in 10 dogs using two wooden boxes. On one side of the boxes, there was a transparent Plexiglas window with small holes that allowed subjects to see or smell the food if they had not witnessed the baiting. The dogs could look and/or smell through the window to check which box the food was in. On the opposite side of the boxes was a lever that subjects had to press to indicate their choice for one of the boxes. In half of the trials, subjects witnessed which box the food was placed into, whereas in the other half of the trials, the subjects’ view of the boxes was obstructed so that baiting could not be witnessed. The dogs selected the correct box with greater accuracy if they had witnessed the baiting process but were at chance levels if they were prevented from seeing the location of the food. The dogs rarely used the opportunity to check the contents of the box before making their choice and did not show a difference in checking behavior between the unseen and the seen condition. This raises the possibility that dogs do not have access to their own perceptual or knowledge states. 

Similarly, McMahon, Macpherson, and Roberts [24] trained dogs to choose from four different boxes, each with a food tray under it. The boxes were all black, except for one box that had a white side, and food was always placed only under this box. Once subjects learned to choose the white sided box, the boxes were rotated 45°, 90°, and 135° on successive sessions. In half of the trials, subjects witnessed which box the food was placed under, whereas in the other half of the trials, the boxes were baited behind a barrier so that the subjects could not witness which box was being baited. Experimenters then noted whether the dogs sought information about the location of the white box by walking around the boxes to a position where they could see the white cue. None of the dogs re-oriented themselves to be able to see the white box. These findings are consistent with those of Bräuer et al. [23]. However, in a second experiment, dogs were trained to approach experimenters and to approach and push aside boxes to obtain a reward hidden underneath. During testing, dogs had to choose among three black boxes with food under only one of them. Before choosing a box, the dogs had to approach one of two people. One person was an informant who subsequently pointed to the correct box. The other person was a noninformant who stood with their back to the dogs while they made their choice. Dogs chose the informant on 65% of the trials and, when choosing the informant, successfully chose the baited box on 73% of the trials. These findings could suggest that dogs will seek information about the correct location of food when that information comes in the form of human cueing but could also be explained by the association of a correct response with the informed experimenter. 

Belger and Brauer [25] reference the importance of allowing animals a natural behavioral response in contrast to an unnatural and overtrained response in order to search for reward. In their study, with a larger sample of domestic dogs, dogs were able to simply look through a gap in fencing to seek information, and to travel around the fencing to obtain rewards. Dogs were more likely to look through the gap on unseen versus seen trials and were more successful on seen trials. These authors concluded that dogs, like most species, are likely capable of responding to uncertainty when provided with a naturalistic task.

Due to the conflicting results obtained when testing canids with an information-seeking paradigm, further evidence is needed to ascertain whether canids exhibit metacognitive capabilities. To date, only domestic dogs have been studied. However, a wealth of evidence suggests that domesticated animals exhibit physical, behavioral, and cognitive differences from their closest wild relatives [26,27,28]. Compared to natural environments, captive environments are more stable and predictable [26]. For example, domestic animals have a low likelihood of being attacked by a (non-human) predator and resources often appear regularly at the same time and location and are easily accessible [26], whereas wild animals often must compete for limited resources and are in danger of predation. It is possible that the history of domestication has diminished dogs’ abilities to cope with the physical environment, with human intervention reducing selective pressures on species-specific behavioral patterns [26]. 

Testing wild canids may benefit our understanding of whether canids have any components of metacognition, and, more importantly, under which conditions metacognition may have evolved. For example, metacognition may have evolved because it provided useful strategies for remembering past actions and linking these to future outcomes. Animals that cache food must remember the locations of their caches and whether they have previously visited and emptied caches [29,30]. One might expect metacognitive abilities in animals such as arctic foxes that cache food, because if animals can monitor the strength of their memory for cache locations, it would allow them to prioritize visiting caches associated with a stronger memory trace [31]. 

Previous studies of metacognition have focused on social species. However, the current studies extend the research on metacognition to relatively less social carnivores by examining whether arctic foxes possess metacognitive capabilities using an information-seeking paradigm. We hypothesized that the subject would immediately choose the baited compartment of an apparatus when he had witnessed the baiting process and would seek information by looking through the transparent part of the apparatus before making a choice when he did not witness the baiting. The subject was also predicted to exhibit longer latencies to choose when he had not witnessed the baiting of the apparatus, as an implicit measure of uncertainty. We did not observe differences in looking behavior in the seen versus unseen baiting conditions across two experiments; thus, we did not obtain evidence of metacognition in the one artic fox that was tested. However, our studies help to inform design of future studies to test metacognition in a species that has previously been absent from the literature.


**Experiment 1**


## 2. Materials and Methods 

### 2.1. Subject

One male arctic fox, Burton, approximately 13 years old, and housed at The Creature Conservancy sanctuary in Ann Arbor, Michigan, was tested during the fall and winter of the period of 2017–2018. Burton was tested in his naturalistic home enclosure, which he shared with a female arctic fox. He was hand-raised and kept as a pet before being surrendered to The Creature Conservancy for further care. He had never participated in an experiment until the current study. Attempts were made to train the female, Mieheira, but she did not respond in the presence of experimenters. Testing was done when there were no visitors to the sanctuary, but other staff members may have been present in the area surrounding the fox enclosure.

### 2.2. Materials

A 45.72 cm × 31.75 cm × 11.43 cm wooden box was used. The box was separated into two equal compartments that could each be opened by pressing down on plastic tabs located at the front of the box as seen in Figure 1A. When depressed, the bottom of the box, which was hinged, opened to release the food reward placed inside. The box also contained two 8.89 cm × 8.89 cm × 3.81 cm Tupperware containers (one in each compartment) that were baited to control for scent cues as seen in Figure 1B. The top of the box was covered with removable transparent plastic so that the subject could see, but not access, the contents of the compartments. A rope was attached to the inside of each compartment and threaded through the back of the box as seen in Figure 1C. These ropes were used to secure the compartment not chosen by the subject during the knowledge test and testing trials. When the experimenter pulled the rope taut from behind the fence, the box could not be opened. The box was hung from the fence of the subject’s enclosure using two ropes (one on each side) and four carabineers (two attached to the box and two attached to the fence) in such a way that the subject needed to rear up on his hind legs in order to view the contents of each compartment through the transparent top, which he should only need to do if he had not seen the baiting. The box compartments, as well as the scent cue containers, were baited with pieces of dog kibble covered in peanut butter and corn meal such that both compartments would smell like the food reward. Therefore, scent alone should not have allowed the subject to choose correctly. A 488 cm x 66 cm metal gate was used to separate the subject from the testing area until a trial began as seen in Figure 1D. Sessions for all phases of the study were recorded using a GoPro Hero 4.

### 2.3. Procedure

Burton was tested twice a week from Oct., 2017 to Feb., 2018 with one–two sessions occurring on each test day. Figure 2 shows the flow of the experiments.

#### 2.3.1. Training

Training was conducted to allow the subject to learn how to access the food reward and to understand that only one box contained a reward on each trial. During all testing, two experimenters were present (henceforth referred to as E1 and E2, with E1 being PB and E2 being TE). During training sessions, the box was secured to the fence inside the subject’s enclosure and the transparent cover from the top of the box was removed. E2 filmed the trials and indicated where food should be placed on each trial from behind the fence. E2 also lifted the box by pulling on the ropes so that the subject did not have access between trials. A trial began with E2 lowering the boxes so that E1, who was inside the enclosure, could bait both compartments, but not the scent cue containers. After the compartments were baited, the cover was placed back on the top of the box by E1 and the subject was able to explore the box and open the compartments. E2 faced away from the subject during the trial. E1 looked toward the center of the box. Training was conducted on two mornings per week before the subject had received his morning ration of food. Each training session consisted of six trials, with each compartment baited three times in a semi-random order. Each trial lasted until the subject had opened both compartments or until he had wandered away from the box for over one minute. Once able to open and retrieve treats from both of the compartments on all trials for two consecutive sessions, the subject progressed to knowledge testing. 

#### 2.3.2. Knowledge Testing

The subject participated in five 6-trial sessions of knowledge testing to ensure that he could accurately track the location of the bait when he had witnessed the baiting process and to teach the subject that he could select only one compartment. The transparent top of the box was removed and both inaccessible scent cue containers were baited to control for scent cues, out of the subject’s view. The box was secured to the fence inside the subject’s enclosure. A metal barrier (visible in Figure 1D) was placed around E1 and the box to keep the subject from interfering with the baiting process. The subject could see the baiting process through the barrier.

Once the barrier was in place, E1 got the subject’s attention by calling the subject’s name and showing him the bait. When E1 had the subject’s attention, she placed an open hand in each compartment at the same time. E1 then removed her hands and placed the transparent cover on the top of the box. The gate was then opened, and the subject was able to explore the box and select a compartment to open. The subject was considered to have made a choice when he made physical contact (using either his mouth or paw) with the tabs on the front of the box. Depressing the tab caused the bottom of the box to drop, releasing the food reward. Once a choice was made, E2 pulled on the rope connected to the compartment not chosen, effectively locking that compartment. Once the subject left the testing area, the gate was closed, and a new trial started. Once the subject was able to accurately select the baited compartment on four out of six trials for two consecutive sessions, he moved into testing. 

#### 2.3.3. Testing

The subject completed six 8-trial test sessions, for a total of 48 completed trials. Each test session consisted of four seen and four unseen trials (described below) presented in random order. Testing sessions took place on two mornings per week before he had received his morning ration of food.

*Seen Trials*. Seen trials followed the same procedure as the knowledge test. After each trial, E2 recorded which compartment the subject selected to open and whether the subject looked through the top of the box before selecting that compartment. E2 also recorded whether the subject sniffed the compartments before making a choice and how long it took him to make a choice. Latency to make a choice was measured from the time the subject’s head passed through the gate to the time the subject made physical contact (using either his mouth or paw) with one of the tabs on the front of the box. Each compartment was baited with food an equal number of times in random order to control for subjects learning a rule such as “left compartment = food.” 

*Unseen Trials*. On unseen trials, the procedure was the same as in the seen trials; however, the subject was never shown which compartment was baited. Instead, E1 baited the box by putting closed fists into each compartment at the same time. In order to view the contents of each compartment, if he chose to do so, the subject needed to stand on his hind legs and look through the top of the box. E2 wrote down whether the subject accurately chose the baited compartment and whether the subject looked through the top of the box before selecting a compartment to open. E2 also wrote down whether the subject sniffed the compartments before making a choice and how long it took the subject to choose a compartment. As in the seen trials, each compartment was baited with food an equal number of times in random order. 

### 2.4. Reliability

A coder not involved in the experiment coded testing trials from video for four of the six test sessions. For two of the test sessions, video was inadvertently not recorded, and the experimenter’s recorded responses were used in analyses. Cohen’s Kappa was calculated to determine the level of agreement between the coder and the experimenter on the accuracy of the subject’s choices and whether the subject looked through the top of the apparatus before making a choice for the four sessions that were videotaped. The coder agreed with the experimenter on the accuracy of the subject’s choices on 100% of these trials (κ = 1.00). For looking behavior, the coder agreed with the experimenter on 93.75% of trials (κ = undefined). In case of disagreement, the coder’s data was used.

Ethical Statement: The animal protocols used in this work were evaluated and waived by the Institutional Animal Use and Ethic Committee (IACUC) of Oakland University.

## 3. Results

### 3.1. Training

The subject reached criterion after five sessions (30 trials).

### 3.2. Knowledge Test 

The subject met criterion to move to testing after completing five sessions (30 trials) of the knowledge test. Across all trials, he performed at 73% accuracy.

### 3.3. Testing 

Analyses were conducted using SPSS v 25 (IBM, Armonk, NY, USA). A binomial test indicated that the observed proportion of the subject accurately selecting the baited compartment when he had witnessed baiting (M = 0.71, SD = 0.46) approached significance (n = 24, *p* = 0.064). The subject’s observed proportion of accurately selecting the baited compartment when he had not witnessed the baiting (M = 0.71, SD = 0.46) also approached significance (n = 24, *p* = 0.064). A Wilcoxon signed-ranks test confirmed that the subject’s choice accuracy during seen trials was equal to his choice accuracy on unseen trials (z = 0.000, *p* = 1.00). See Figure 3.

Furthermore, there was no significant difference between the subject’s looking behavior during seen and unseen trials (z = −1.41, *p* = 0.157). In fact, the subject looked through the transparent top only twice, both times during seen trials. The subject’s latency to make a choice when he had witnessed baiting (M = 6.42, SD = 5.15) was not significantly different from his latency to make a choice when he had not witnessed baiting (M = 6.5, SD = 8.2, Z = −0.414, *p* = 0.679). See Figure 4. 

## 4. Discussion

Burton passed the knowledge test in relatively few trials, indicating that he was able to attend to which compartment was baited and to use such information to guide his search. We attempted to eliminate the use of scent cues by baiting both compartments with an inaccessible reward so that the subject had to rely on visual cues to find the reward. His performance in this phase suggested that his vision was adequate for progressing to the testing phase of the experiment.

To infer that an animal exhibits metacognition using the information-seeking paradigm, the animal must accurately select the baited container when it has witnessed the baiting process and must actively seek out the missing information before selecting a compartment when it has not witnessed the baiting process. When tasked with an information-seeking paradigm, animals that do not exhibit metacognition randomly choose a compartment to secure a reward or will seek information even when witnessing the baiting process. The results of this experiment show that the subject accurately selected the baited compartment both when he had and had not witnessed the baiting process. However, the subject did not take the opportunity to look through the transparent top of the apparatus to identify which compartment was baited selectively when he had not witnessed the baiting process. It is unlikely that it was difficult for him to engage in this behavior because he did so twice; however, he did so when it was unnecessary because he had witnessed the baiting on both of those trials. These findings, with the exception of high choice accuracy on unseen trials, are consistent with the findings of some previous studies conducted with domestic dogs in which subjects did not engage in information-seeking behavior (e.g., [23,24]).

High levels of accuracy on unseen trials suggest that other cues were available to assist the subject in making accurate choices despite efforts to reduce visual and scent cues. Possibly, the scent cues used were ineffective or the subject was able to view the bait through small gaps in the wood near the bottom of each compartment. Both experimenters knew which compartment was baited on each trial, and although E2 looked away and E1 standardized her gaze during the trials, it is possible that she provided inadvertent cues to the subject. In addition, the subject might not have been motivated to seek information due to the high probability of accurately selecting the baited container simply by guessing (given only two options). To resolve these potential issues, a new apparatus that better controlled for scent and visual cues, and which contained a third compartment to lower the probability of selecting the baited compartment by guessing alone was used in Experiment 2. 


**Experiment 2**


## 5. Materials and Method

### 5.1. Subjects

The same male arctic fox that was tested in Experiment 1 was tested again in the same enclosure.

### 5.2. Materials

Three Sterilite 9.8 L step-on wastebaskets were used, as seen in Figure 5. The backs of the wastebaskets were cut out and replaced with transparent pieces of Plexiglas, creating a window in the wastebasket. Shelves made out of thick cardboard were placed inside the wastebaskets to make the wastebaskets shallower, thereby making the bait easier to access. The wastebaskets were affixed to a piece of plywood and were spaced 36 cm apart. Multi-purpose adjustable lock straps were attached to each wastebasket to secure the lids of the wastebaskets not chosen by the subject during the knowledge test and experimental trials. The wastebaskets were baited with pieces of dog kibble covered in peanut butter and corn meal. The wastebaskets contained a smaller, removable wastebasket housed in a larger container. To control for scent cues, the smaller wastebasket was removed, and a piece of bait was placed on the bottom of the housing container. The smaller wastebasket was then reinserted, hiding the bait from sight. A 66 cm x 488 cm metal gate was used to separate the subject from the testing area until a trial began. Sessions for all phases of the study were recorded using a GoPro Hero 4.

### 5.3. Procedure

#### 5.3.1. Training 

The apparatus was placed in Burton’s enclosure overnight for several days prior to training to familiarize him with the fact that there were transparent windows at the back of the wastebaskets. Training provided the opportunity for the subject to explore the wastebaskets and see that the back was transparent (see Figure 5B) and learn how to open the lids of the wastebaskets. The subject could either step on the paddle that popped the lid open or push the lid up and open using his nose or paw against the lip of the lid. During training sessions, the wastebaskets were placed in the subject’s enclosure. Each wastebasket was baited. Once the lids of the wastebaskets were closed, the subject was able to explore the wastebaskets and open the lids. Training was conducted on two mornings per week before the subject had received his morning ration of food. Each training session consisted of six trials, with each compartment baited twice in a semi-random order across trials by E1. Once able to open and retrieve treats from all three wastebaskets on all trials for two consecutive sessions, the subject progressed to knowledge testing. 

#### 5.3.2. Knowledge Testing

Knowledge testing was conducted to ensure that the subject could accurately track the location of the bait when he had witnessed the baiting process and to teach the subject that he could select only one compartment. Sessions consisted of six trials. The subject was tested on two mornings per week, completing one session a day. During knowledge testing, the same two experimenters were present. The wastebaskets were placed in the subject’s enclosure in such a way that the windows were not initially visible to the subject, but he could walk around behind them to see the windows. A metal gate was placed around E1 and the wastebaskets to keep the subject from interfering with the baiting process. Once the gate was in place, E1 placed bait in the bottom of each wastebasket, out of the subject’s view, to control for scent cues. E1 then got the subject’s attention by calling the subject’s name and showing him the bait. When E1 had the subject’s attention, she placed an open hand in the wastebasket that was being baited. E1 then closed the lids of the wastebaskets. The lids were always opened and closed in the same order to prevent the subject learning a rule such as “first lid opened = reward.” Once the lids were closed, E1 opened the gate and the subject was able to explore the wastebaskets and choose a lid to open. The subject was considered to have made a choice when he made physical contact (using either his mouth or paw) with the wastebasket. Once a choice was made, E1 sealed the lids of the wastebaskets not chosen using the adjustable lock straps. Once the subject left the testing area, the gate was closed, and a new trial started. Once the subject accurately selected the baited compartment on four out of six trials for two consecutive sessions, he moved into testing. 

#### 5.3.3. Testing 

The order of presentation of seen and unseen trials was randomized such that the subject received three trials of each type in a semi-random order for a total of six test trials per session. The subject completed two test sessions, for a total of 12 completed trials. Testing took place on two mornings per week before the subject had received his morning ration of food.

##### Seen Trails

Seen trials followed the same procedure as the knowledge test. After each trial, E2 recorded which wastebasket the subject selected to open and whether he looked through the back of the wastebaskets before selecting a wastebasket to open. E2 also recorded whether the subject sniffed the wastebaskets before making a choice and latency to make a choice, which was recorded as the time the subject made physical contact (using either his mouth or paw) with a wastebasket from the time his head passed through the gate. Each wastebasket was baited with food an equal number of times in random order. 

##### Unseen Trails 

On unseen trials, the procedure was the same as in the seen trials; however, the subject was never shown which wastebasket was baited. Instead, E1 baited the wastebasket out of sight of the subject. In order to view the contents of each wastebasket, the subject had to travel to the back of the wastebaskets and look through the windows. E2 made a note of whether the subject accurately chose the baited wastebasket and whether the subject looked through the windows in the back of the wastebaskets before selecting a wastebasket to open. E2 also recorded whether the subject sniffed the wastebaskets before making a choice and latency to choose a wastebasket. As in the seen trials, each compartment was baited with food an equal number of times in random order.

### 5.4. Reliability

A coder coded testing trials from video for both test sessions. The coder agreed with the experimenter on the accuracy of the subject’s choices on 100% of these trials (κ = 1.00). For looking behavior, the coder agreed with the experimenter on 100% of trials (κ = 1.00).

## 6. Results

### 6.1. Training

The subject reached criterion after two sessions (12 trials).

### 6.2. Knowledge Test

The subject reached criterion to move to testing after completing four sessions (24 trials). Across all trials, he performed at 54% accuracy.

### 6.3. Testing

A binomial test indicated that the observed proportion of accurate choices on seen trials (M = 0.33, SD = 0.52) was not significantly different from the expected probability given chance rates (n = 6, *p* = 0.687). The same was true of unseen trials (M = 0.33, SD = 0.52, n = 6, *p* = 0.687). A Wilcoxon signed-ranks test confirmed that the subject’s choice accuracy during seen trials was equal to his choice accuracy on unseen trials (*z* = 0.00, *p* = 1.00). See Figure 6. 

Furthermore, there was no significant difference between the subject’s looking behavior during seen and unseen trials (*z* = 0.00, *p* = 1.00). In fact, the subject never took the opportunity to look through the window in the back of the wastebaskets. The subject’s latency to make a choice when he had witnessed baiting (M = 13.06, SD = 14.7) was actually slower than his latency to make a choice when he had not witnessed baiting (M = 5.9, SD = 1.54), although this difference was not statistically significant (*z* = −0.734, *p* = 0.463). See Figure 7.

## 7. Discussion

Most critically, Burton never took the opportunity to look through the transparent section of the apparatus during seen or unseen trials. This finding is consistent with previous tests of metacognition in domestic dogs [23,24] that required an operant response, such as pushing a lever, but not consistent with [25], in which dogs could simply look through a gap before travelling past it to obtain the reward. Burton did not seek information when he should have been uncertain. Unlike in Experiment 1, when he might have not needed to seek information because he was able to find the bait accurately without doing so, he would have benefitted from engaging in information seeking with the apparatus used in Experiment 2 based on his low accuracy across testing trials.

Prior work with domestic dogs has shown that dogs accurately select a baited container at above chance levels when they witnessed the baiting of the container, but at or below chance levels when they did not witness the baiting of the container [23,24]. Although in one study, dogs were above chance on unseen trials as well, dogs also utilized the option to look for helpful information in that study [25]. Although Burton had passed criterion during knowledge testing, his accuracy was below chance levels for both seen and unseen trials during testing, which suggests challenges with the testing situation. The experimenter did her best to get the subject’s attention and to bait the compartments when the subject was looking; however, it is possible that the subject was not paying attention during the baiting process and this could account for his low choice accuracy on seen trials. It is also possible that the subject’s declining health influenced the results of the study. Although he was motivated to participate even during the final test sessions, Burton was experiencing pain and reduced mobility due to Intervertebral Disc Disease during Experiment 2. Unfortunately, he had to be euthanized after having completed only two test sessions. 

## 8. General Discussion

Metacognition involves the monitoring and control of one’s own cognition. It was once thought that only humans were capable of metacognition. However, evidence from recent decades using both the uncertainty response paradigm and the information-seeking paradigm suggests that some animals, such as dolphins [3], rats [4], rhesus macaques [5,32], chimpanzees, orangutans [6], and western scrub jays [7], may possess metacognitive abilities. Results from tests with domestic dogs have been mixed [23,24,25]. The aim of the studies reported in this paper was to extend the findings on metacognition in canids by testing a wild canid species. The arctic fox is a good candidate for such studies, given that foxes, like scrub jays, engage in caching behavior and may need to remember the location, contents and timing of caches [33]. Furthermore, researchers have focused on social species, neglecting less social species that may engage in metacognition in order to plan and remember foraging activities. We argue that more experiments should be conducted with species based on foraging ecologies, rather than sociality or relatedness to humans. Although results from our 13-year-old arctic fox did not suggest the use of metacognition to inform choice behavior, we believe that we have demonstrated a low-tech method for assessing these abilities in similar species. Clearly more individuals need to be tested, particularly those that may have experienced more typical living conditions for wild foxes. It may be that captive foxes do not demonstrate the ability to keep track of what they have seen, given that they may no longer need to cache. Although the individual that we tested did sometimes cache his food and was housed with a conspecific that might have exploited his caches, we cannot know how essential this life experience is to the caching behavior and metacognitive capabilities of foxes without testing larger samples with diverse rearing histories. Attending to individual differences is particularly important when basing conclusions about species capacities based on very small samples [34].

Our procedure allowed several tests of fox cognition. First, although not the goal of the study, we were able to demonstrate that the fox could accurately retrieve bait when he had witnessed it being placed within an apparatus (in Experiment 1 and on knowledge trials of Experiment 2), suggesting some degree of object permanence and/or short-term memory. Given the lack of work on memory and retention in canids, we suggest that these are fruitful areas for future investigations. Second, and more pertinent to the goal of the study, we were able to assess both implicit and explicit measures of metacognition. We predicted that, if the subject implicitly recognized that he knew where the bait was located only on seen trials, he should respond more quickly on those trials relative to unseen trials. We did not observe such a finding. Although unexpected, this result is consistent with data from domestic dogs [25], who also responded just as quickly when they did not have information about where food was baited as when they did. It is possible that dogs are not sensitive to information about where food has been placed if it is relatively low effort and effective to search widely [25]. Future studies should impose more of a cost on searching indiscriminately by introducing competition or increasing effort to find rewards. It is also possible that poor inhibition affects responses of canids in these paradigms.

The explicit measure of metacognition was whether the subject selectively sought available information about where the bait was hidden on unseen trials. He sought this information only twice—both times in Experiment 1 and both times on seen trials. The failure to exhibit selective seeking behavior in Experiment 1 may have been due to the fact that the subject did not need to engage in such behavior to accurately recover the bait. He accurately selected the baited compartment of the apparatus at above chance levels both when he had and had not witnessed baiting, which could have been due to ineffective blocking of scent cues, visual information gathered through small gaps in the wood near the bottom of each compartment, or a combination of the two. Further, the subject may not have been motivated to seek information due to the high probability of accurately selecting the baited container simply by guessing given that there were only two locations where bait might be recovered. Dogs also retrieved bait at above chance levels on unseen trials in a previous study [25], but presumably this was due to engaging in seeking behavior prior to making choices of where to search.

For Experiment 2, a new apparatus was built to better control for the issues in Experiment 1. In this experiment, Burton’s choice accuracy was below chance for both seen and unseen trials. These results may have been influenced by the subject’s rapidly declining health. Based on these experiments alone, we cannot suggest that arctic foxes do not have access to their own perceptual or knowledge states. Future studies with more subjects are needed to better determine whether arctic foxes possess metacognitive capabilities. Furthermore, it is important to design more naturalistic studies that take advantage of the foxes’ natural caching behavior [33]. Having foxes witness conspecifics bury food rather than having them attend to human behavior may be more likely to result in affirmative results. Additionally, our task was based upon a paradigm that asks subjects whether they know what they have seen. A better suited test for canids, and other carnivores, might require them to reflect upon what they have smelled or heard instead of what they have seen. Of course, such tasks are more difficult to design, as it is more difficult to control and determine the dispersal of olfactory and auditory cues. It is also difficult to create a task in which a subject can take an action to provide themselves with auditory or olfactory information, whereas it is relatively simple to have a subject change orientation or position to gain visual access. Although one can imagine a test in which a vented area is available that a subject must sniff through, rather than to look through, one would have to be certain that the scent cues were not available without taking such an action. 

## 9. Conclusions

Our work suggests that a simple methodology can be effective for testing wild canids, provided visual and olfactory cues are well controlled for. Additional research is needed to determine whether arctic foxes, canids, and, more broadly, carnivores are capable of metacognition. Researchers should not focus on highly social species alone but should compare species that differ in other important ways, such as the extent to which they engage in extractive foraging or caching. Testing wild caught individuals in addition to captive individuals would also be informative.

## Figures and Tables

**Figure 1 behavsci-10-00081-f001:**
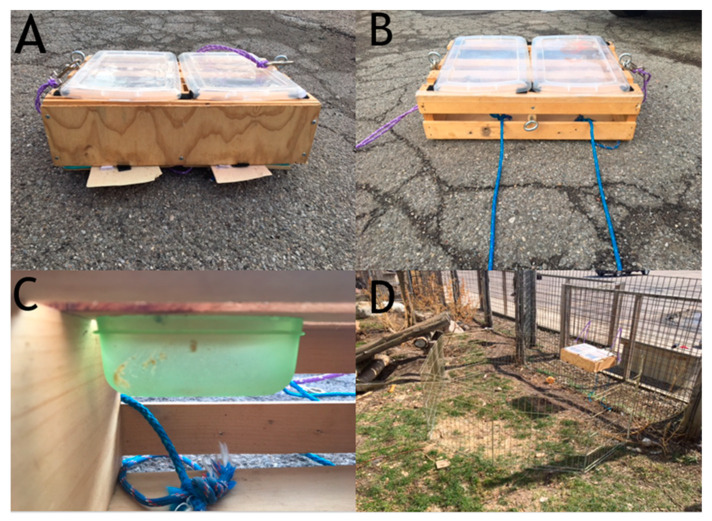
The apparatus used for testing in Experiment 1. (**A**) Front of the apparatus showing tabs used for opening compartments. (**B**) Back of the apparatus showing ropes used to lock the compartment that was not chosen. (**C**) Inside of the apparatus showing scent cue containers. (**D**) Testing setup. E2 stood behind the fence visible in D and E1 stood outside the temporary gate area inside the enclosure.

**Figure 2 behavsci-10-00081-f002:**
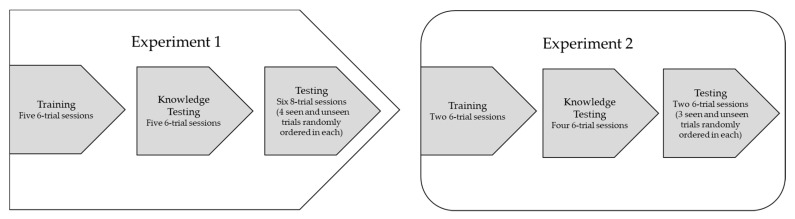
The flow of the experiments. Each phase lasted 2–3 weeks.

**Figure 3 behavsci-10-00081-f003:**
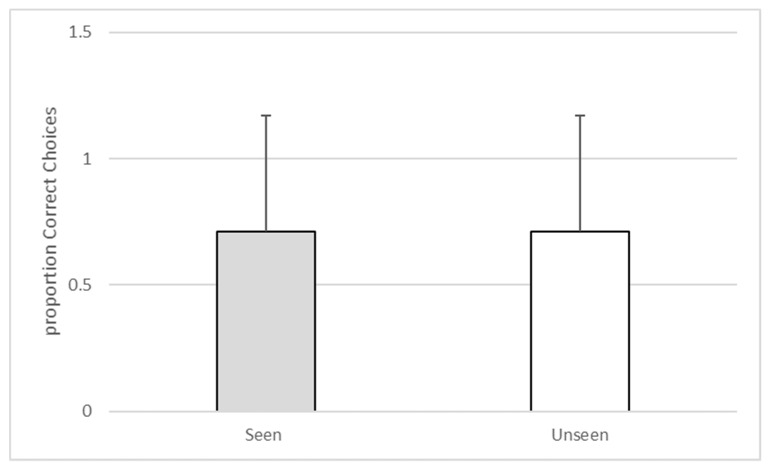
The proportion of correct choices during seen versus unseen trials during Experiment 1.

**Figure 4 behavsci-10-00081-f004:**
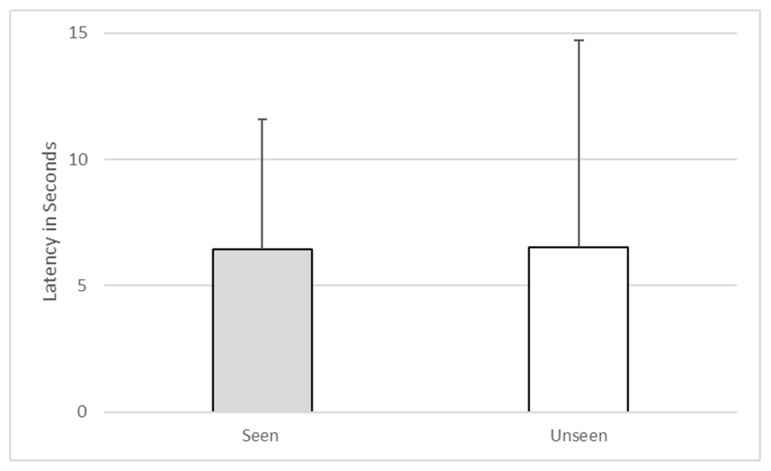
Latency to make a choice on seen versus unseen trials during Experiment 1.

**Figure 5 behavsci-10-00081-f005:**
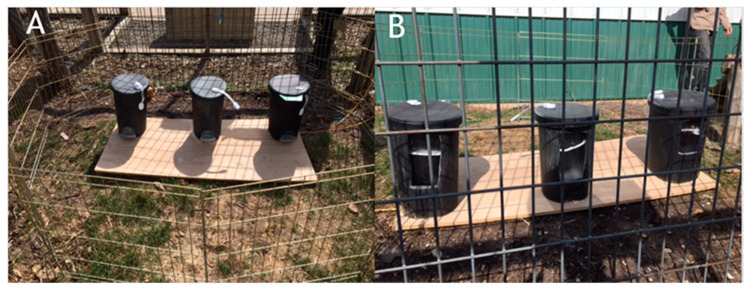
The apparatus and set up used for testing in Experiment 2. (**A**) Front of the apparatus. (**B**) Back of the apparatus.

**Figure 6 behavsci-10-00081-f006:**
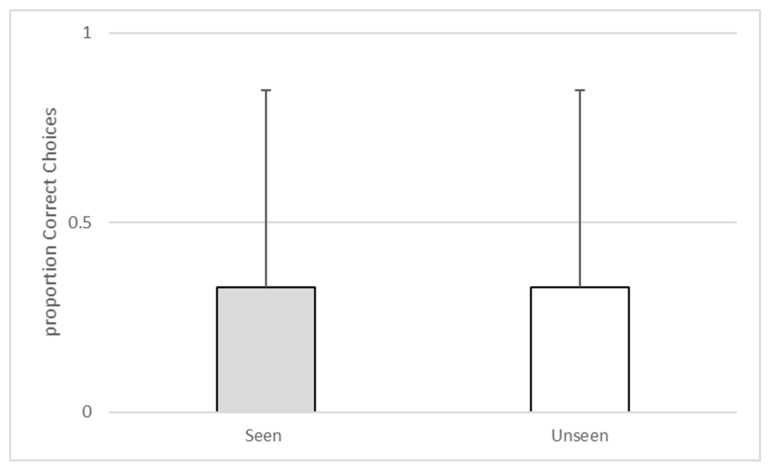
The proportion of correct choices during seen versus unseen trials during Experiment 2.

**Figure 7 behavsci-10-00081-f007:**
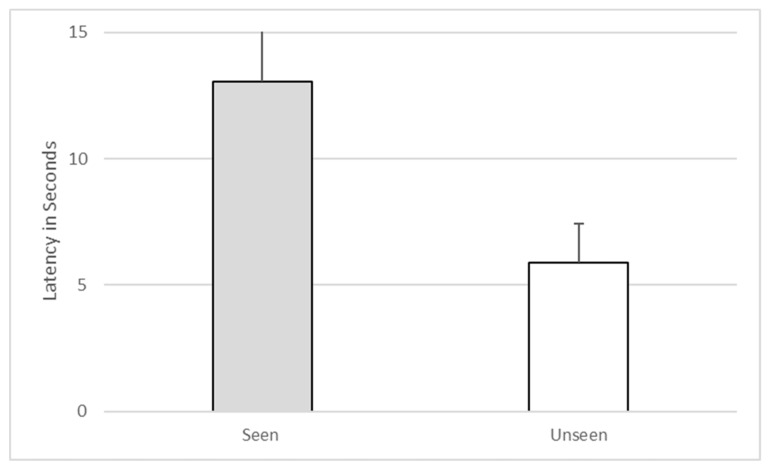
Latency to make a choice on seen versus unseen trials during Experiment 2.

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
