# Peer review of "Are there Metacognitivists in the Fox Hole? A Preliminary Test of Information Seeking in an Arctic Fox (Vulpes lagopus)"

_behavsci, 2020, doi:10.3390/bs10050081_

Round 1

Reviewer 1 Report

this is an interesting paper and overall is well written.

I would like to see further clarity in the methods, particularly experiment one

The rationale for requiring fox to have to stand on hind legs to look into the top of the box, and only it seems for the unseen trials is not clear. 

of itself it may have been a confounding variable, given age of animal (potential exacerbation of arthritic pain when standing up, eyesight problems.. neither tested for).

the actual overall  method section could be improved by timelines / diagram

Discussion : your discussion is overall fine, but I would like to see more emphasis on the need to consider the individual both in terms of prior experiences and sensory abilities. for example not needing to cache food in captivity.. or it not being a reinforced behaviour if caches are cleared away by keepers... it will extinguish..so the individual may not 'learn' to know what it knows.

likewise consideration of sensory abilities must be not only at species level but also at individual level.. a younger animal for example may well have given different results in exp 2.. but still may have been based on scent detection and 'quantity' difference with the scent in the baited compartment being stronger... think of arctic fox scent abilities of finding caches under snow for example, or a medical detection dog (or rat's) abilities....

simply knowing my own dog's decline in scenting and tracking and free searching . still good at 13.5 years, it is noticeably inferior to even a year ago, as is eye sight.

of course these reductions in sensory ability, physical low level chronic pain can affect animals at all ages.. and as researchers we need to be much more cognisant of this, as well as of individual learning (informal or formally trained) experiences...an issue for ALL studies of cognition and metacognition.

this paper provides a good space to really emphasise these confounding variables.. we need to ask animals questions in ways they can answer... and that means taking into account species (and breed) and individual variables. 

Author Response

this is an interesting paper and overall is well written.

 Thank you for your positive comments.

I would like to see further clarity in the methods, particularly experiment one

We have added some detail to the Methods sections.

The rationale for requiring fox to have to stand on hind legs to look into the top of the box, and only it seems for the unseen trials is not clear. 

of itself it may have been a confounding variable, given age of animal (potential exacerbation of arthritic pain when standing up, eyesight problems.. neither tested for).

The fox was not required to stand up on his hind legs per se. However, this was a means by which he could check if he could see the bait before making a choice specifically on trials where he had not already seen the baiting process. We have added some wording here to clarify. It is true that the procedure in Exp. 1 may have made it less likely that Burton would engage in looking behavior given arthritic pain but we observed that he reared up to look around his habitat outside of testing without any sign of pain. Furthermore, he did look twice during the six testing trials, but he did so during seen rather than unseen trials, when there was no need for him to do so. We have clarified and added to the discussion on pg. 9-10. The main problem in Exp. 1 was that he did not need to look because he was equally successfully in “guessing” correctly on unseen trials anyway.

the actual overall method section could be improved by timelines / diagram

 We have inserted a flowchart depicting the flow of both experiments with a timeline directly under the Procedure heading. Please advise if you wish us to insert elsewhere.

Discussion : your discussion is overall fine, but I would like to see more emphasis on the need to consider the individual both in terms of prior experiences and sensory abilities. for example not needing to cache food in captivity.. or it not being a reinforced behaviour if caches are cleared away by keepers... it will extinguish..so the individual may not 'learn' to know what it knows.

These are helpful points and we have added discussion on lines 529-535 to reflect possible individual reasons for performance.

likewise consideration of sensory abilities must be not only at species level but also at individual level.. a younger animal for example may well have given different results in exp 2.. but still may have been based on scent detection and 'quantity' difference with the scent in the baited compartment being stronger... think of arctic fox scent abilities of finding caches under snow for example, or a medical detection dog (or rat's) abilities....

simply knowing my own dog's decline in scenting and tracking and free searching . still good at 13.5 years, it is noticeably inferior to even a year ago, as is eye sight.

of course these reductions in sensory ability, physical low level chronic pain can affect animals at all ages.. and as researchers we need to be much more cognisant of this, as well as of individual learning (informal or formally trained) experiences...an issue for ALL studies of cognition and metacognition.

this paper provides a good space to really emphasise these confounding variables.. we need to ask animals questions in ways they can answer... and that means taking into account species (and breed) and individual variables. 

I completely agree with the reviewer on these points. They are important considerations, particularly with dealing with such small samples, as, in our case, a single subject. We have added more discussion on these points on lines 559-564 and 366-370. However, I would like to note that Burton’s success in Exp. 1 likely indicates adequate visual acuity as well as potentially a strong sense of smell that allowed him to succeed on the unseen trials despite our attempt to control for scent. The fact that he reared up twice to look into the compartments but did not do so selectively on unseen trials suggests that his lack of looking on unseen trials was not due to an inability to do so (lines 390-392).

Reviewer 2 Report

This manuscript presents single-subject data related to the question of whether an arctic fox would demonstrate evidence of metacognitive abilities. The authors findings were negative, which does not mean that foxes in general are not capable of metacognition (absence of evidence is not evidence of absence) and I agree with their assertion that foxes would be good candidates for further testing of this hypothesis. I would be very pleased to see such negative findings published. I would also like to commend the authors on a well written manuscript, which I enjoyed reading (…until the part where the fox died).

Line 35: full stop after [6] should be a comma, and a closed parenthesis is needed after [8,9].

Line 36: as this journal will be read by human psychologists who may be less familiar with animal behaviour, should an explanation be given for what caching involves? e.g. ‘Given that arctic foxes cache their food for potential future need, which involves remembering cached locations…’ just a suggestion, the authors do not have to do this unless they think it worthwhile.

Line 145: I think this bit is meant to mean: if the dogs were prevented from seeing the boxes being baited; as they could see the location of the food if they wanted to. I think however that there is another issue here with testing dogs, which is that of domestication. Studies looking at deception in pointing tests show that dogs appear to value our human guidance over information provided to them by their own senses. The willingness to follow human cues over their own judgements (when they can see which location is baited) seems to be impacted by breed selection for cooperative traits (e.g. Barnard et al 2019 https://link.springer.com/article/10.1007/s10071-019-01272-3). Essentially, dogs may have access to their own perceptual knowledge, but may be more inclined to inhibit their use of it. Some dogs have been shown to ignore a deceptive human gesture in favour of their own knowledge when food is visible (see figure 2 here https://www.researchgate.net/publication/230801007_Domesticated_Dogs'_Canis_familiaris_Response_to_Dishonest_Human_Points), but more than 50% of dogs seem to still choose to follow a deceptive static point, not only inhibiting their response to the sight of food but inhibiting their response to self-knowledge in favour of human social cues. I know this doesn’t relate to whether they will actively seek knowledge, but these studies highlight a potential issue with testing metacognition in dogs. Don’t feel like you have to incorporate this, I just thought it worth mentioning here, so that you can add some discussion on this if you think it worthwhile. Another issue here is that of senses; we primarily gather information by looking for it, whereas for dogs, sight is not so important, and they will rely more upon scent. So, ideally, a test like this for dogs should utilise a scent-based paradigm.

Line 263-4: Did E1 bait both boxes, or just one? I assume from line 270, it was only one, but this isn’t clear here.

Line 267: It’s not clear what happened when Burton pulled on the tabs; I’m presuming he released the baited food somehow, but this isn’t explicit.

Line 269: how much time was Burton given to explore the second box to learn that it wouldn’t now open? This could do with being stated here.

The grouping of Experiment 2 data in the results figures for Experiment 1 is strange. I can see why you did it – so they can be seen together, but for the reader who hasn’t yet come across Exp.2 it is odd, especially as we have to scroll back up a long way when we reach the Exp. 2 Results section. I’d suggest re-jigging your figures so that the Choice and Latency from Exp.1 are shown on the same figure as A and B sections. Then put these figures as they are into Results 2, so that the two sets of data can be compared there. (Having now read Exp 2, the data doesn’t seem to need comparing, so separate figures would make most sense).

Line 439-440: Did Burton know there was a transparent window around the back that he could look through? Without knowing there was an opportunity for gaining knowledge, what this test could really be testing is how willing he is to take a guess versus inhibiting his choice and trying new strategies. Since there’s no real cost to failure, he has no incentive to hold back and try something new, so may as well gamble. I do understand your point about having just 2 choices meaning a 50/50 pay-off, but doesn’t that assume that he can judge probabilities? It seems a bit of s stretch to me to think that he will know that more boxes means lower odds of getting food (relates also to line 506). This doesn’t impact the results at all, but a two-box choice is probably fine unless it can be shown that they understand probability. A future paradigm could impose a cost on getting the choice wrong, something like a time delay perhaps. If both choices are rewarded but one with preferred food immediately and one with least preferred food after a time delay, this could provide a greater incentive to inhibit their choice to gather knowledge. I now see you have briefly alluded to this last point in line 495-7, but I do feel like the General Discussion could be added to with further options for future tests, perhaps including a suggestion to test information gain via scent or auditory modalities.

Line 410: there’s a double space before the last word ‘recorded’

Oh no, poor Burton! I didn’t expect him to die! :(

Author Response

This manuscript presents single-subject data related to the question of whether an arctic fox would demonstrate evidence of metacognitive abilities. The authors findings were negative, which does not mean that foxes in general are not capable of metacognition (absence of evidence is not evidence of absence) and I agree with their assertion that foxes would be good candidates for further testing of this hypothesis. I would be very pleased to see such negative findings published. I would also like to commend the authors on a well written manuscript, which I enjoyed reading (…until the part where the fox died).

 Thank you very much for your positive comments.

Line 35: full stop after [6] should be a comma, and a closed parenthesis is needed after [8,9].

 Thank you for catching these errors, which are now fixed.

Line 36: as this journal will be read by human psychologists who may be less familiar with animal behaviour, should an explanation be given for what caching involves? e.g. ‘Given that arctic foxes cache their food for potential future need, which involves remembering cached locations…’ just a suggestion, the authors do not have to do this unless they think it worthwhile.

 This is a great suggestion. We have reworded as suggested, line 36.

Line 145: I think this bit is meant to mean: if the dogs were prevented from seeing the boxes being baited; as they could see the location of the food if they wanted to.

We have reworded as suggested for clarity on line 146.

 I think however that there is another issue here with testing dogs, which is that of domestication. Studies looking at deception in pointing tests show that dogs appear to value our human guidance over information provided to them by their own senses. The willingness to follow human cues over their own judgements (when they can see which location is baited) seems to be impacted by breed selection for cooperative traits (e.g. Barnard et al 2019 https://link.springer.com/article/10.1007/s10071-019-01272-3). Essentially, dogs may have access to their own perceptual knowledge, but may be more inclined to inhibit their use of it. Some dogs have been shown to ignore a deceptive human gesture in favour of their own knowledge when food is visible (see figure 2 here https://www.researchgate.net/publication/230801007_Domesticated_Dogs'_Canis_familiaris_Response_to_Dishonest_Human_Points), but more than 50% of dogs seem to still choose to follow a deceptive static point, not only inhibiting their response to the sight of food but inhibiting their response to self-knowledge in favour of human social cues. I know this doesn’t relate to whether they will actively seek knowledge, but these studies highlight a potential issue with testing metacognition in dogs. Don’t feel like you have to incorporate this, I just thought it worth mentioning here, so that you can add some discussion on this if you think it worthwhile.

This is a really interesting point. However, we are not sure it would apply to this fox, which is not domesticated, so we have opted to leave out discussion for fear of confusing the issue.

Another issue here is that of senses; we primarily gather information by looking for it, whereas for dogs, sight is not so important, and they will rely more upon scent. So, ideally, a test like this for dogs should utilise a scent-based paradigm.

Yes, we agree with the reviewer. We have added consideration of this point to the discussion on lines 601-610.

Line 263-4: Did E1 bait both boxes, or just one? I assume from line 270, it was only one, but this isn’t clear here.

Both scent cue containers were baited as noted because this means that both boxes could smell like food so that the fox could not use scent alone to choose the correct box. We have clarified with, “both inaccessible scent cue containers were baited to control for scent cues…”

Line 267: It’s not clear what happened when Burton pulled on the tabs; I’m presuming he released the baited food somehow, but this isn’t explicit.

 On lines 292-293, we have added, “Depressing the tab caused the bottom of the box to drop, releasing the food reward.”

Line 269: how much time was Burton given to explore the second box to learn that it wouldn’t now open? This could do with being stated here.

At the beginning of the Training section 2.3.1, we have added, “Training was conducted to allow the subject to learn how to access the food reward and to understand that only one box contained a reward on each trial.” Burton received 30 trials of training.

The grouping of Experiment 2 data in the results figures for Experiment 1 is strange. I can see why you did it – so they can be seen together, but for the reader who hasn’t yet come across Exp.2 it is odd, especially as we have to scroll back up a long way when we reach the Exp. 2 Results section. I’d suggest re-jigging your figures so that the Choice and Latency from Exp.1 are shown on the same figure as A and B sections. Then put these figures as they are into Results 2, so that the two sets of data can be compared there. (Having now read Exp 2, the data doesn’t seem to need comparing, so separate figures would make most sense).

 We have created separate figures for each experiment as suggested.

Line 439-440: Did Burton know there was a transparent window around the back that he could look through? Without knowing there was an opportunity for gaining knowledge, what this test could really be testing is how willing he is to take a guess versus inhibiting his choice and trying new strategies.

We have added information about familiarization to the apparatus on lines 433-435.

Since there’s no real cost to failure, he has no incentive to hold back and try something new, so may as well gamble. I do understand your point about having just 2 choices meaning a 50/50 pay-off, but doesn’t that assume that he can judge probabilities?

We certainly didn’t mean to imply this – just that there was such a high probability of being correct by guessing alone that the hit rate was high enough to not incentivize changing his strategy, as you say.

 It seems a bit of s stretch to me to think that he will know that more boxes means lower odds of getting food (relates also to line 506).

Again, we did not mean to suggest this. But we hoped that if the rate of correct guessing declined, his accuracy would decline and thus, his motivation to attempt other strategies such as selectively seeking other information on unseen trials would be more readily observed.

This doesn’t impact the results at all, but a two-box choice is probably fine unless it can be shown that they understand probability. A future paradigm could impose a cost on getting the choice wrong, something like a time delay perhaps. If both choices are rewarded but one with preferred food immediately and one with least preferred food after a time delay, this could provide a greater incentive to inhibit their choice to gather knowledge. I now see you have briefly alluded to this last point in line 495-7, but I do feel like the General Discussion could be added to with further options for future tests, perhaps including a suggestion to test information gain via scent or auditory modalities.

We have expanded our discussion of methodological considerations as per both reviewers’ comments.

 Line 410: there’s a double space before the last word ‘recorded’

 We have fixed this. Thank you.

Oh no, poor Burton! I didn’t expect him to die! :(

We were extremely upset about his death as well. We had no idea of his illness during testing of the first experiment and were surprised at how quickly his condition declined after the first two sessions of testing. We halted testing once we observed his difficulty in moving even though he was still willing to participate.